# Impact of Perinatal Death on the Social and Family Context of the Parents

**DOI:** 10.3390/ijerph17103421

**Published:** 2020-05-14

**Authors:** Cayetano Fernández-Sola, Marcos Camacho-Ávila, José Manuel Hernández-Padilla, Isabel María Fernández-Medina, Francisca Rosa Jiménez-López, Encarnación Hernández-Sánchez, María Belén Conesa-Ferrer, José Granero-Molina

**Affiliations:** 1Department of Nursing, Physiotherapy and Medicine, University of Almeria, 04120 La Cañada de San Urbano, Spain; isabel_medina@ual.es (I.M.F.-M.); rjimenez@ual.es (F.R.J.-L.); jgranero@ual.es (J.G.-M.); 2Faculty of Health Sciences, Universidad Autónoma de Chile, Temuco 01090, Chile; 3Hospital La Inmaculada, 04600 Huércal-Overa, Spain; marcos_caav@hotmail.com; 4Hospital de Torrevieja, 03186 Torrevieja, Spain; ehsanchez@ucam.edu; 5School of Health and Education, Middlesex University, London NW4 4BH, UK; 6Faculty of Health Sciences, Universidad Católica de San Antonio de Murcia, 30107 Guadalupe de Maciascoque, Spain; 7Faculty of Health Sciences, Universidad de Murcia, 30003 Murcia, Spain; mb.conesaferrer@um.es

**Keywords:** perinatal death, perinatal grief, disenfranchised grief, qualitative research, parents care

## Abstract

Background: Perinatal death (PD) is a painful experience, with physical, psychological and social consequences in families. Each year, there are 2.7 million perinatal deaths in the world and about 2000 in Spain. The aim of this study was to explore, describe and understand the impact of perinatal death on parents’ social and family life. Methods: A qualitative study based on Gadamer’s hermeneutic phenomenology was used. In-depth interviews were conducted with 13 mothers and eight fathers who had suffered a perinatal death. Inductive analysis was used to find themes based on the data. Results: Seven sub-themes emerged, and they were grouped into two main themes: 1) perinatal death affects family dynamics, and 2) the social environment of the parents is severely affected after perinatal death. Conclusions: PD impacts the family dynamics of the parents and their family, social and work environments. Parents perceive that society trivializes their loss and disallows or delegitimizes their grief. Implications: Social care, health and education providers should pay attention to all family members who have suffered a PD. The recognition of the loss within the social and family environment would help the families to cope with their grief.

## 1. Introduction

Perinatal death (PD) occurs between the 22nd full week of gestation (or when the baby weighs 500 g) and 7 days after birth [1], and although it has decreased globally [2], about 2% of pregnancies end in stillbirth [3]. The worldwide PD rate is estimated at around 2.7 million deaths per year [4]. In Europe, the average PD rate is 5.5 deaths for every 1000 births [5] and is lower in Spain with 4.43 deaths for every 1000 births, representing some 2000 affected families each year [6].

PD is one of the most painful experiences for parents [7,8], who have to undergo grieving and coping processes that include biological, psychological, social and spiritual aspects [9,10]. Grief after a PD is a normal and individual process with multiple biopsychosocial repercussions [11]. The physical symptoms include decreased appetite, weight loss [12], insomnia [13], increased chronic diseases, and decreased quality of life [14]. On a psychological level, parents who suffer a PD have a greater predisposition to suffer anxiety, depression [15], post-traumatic stress syndrome [16], and even an increased risk of suicide [17]. In the family environment, the death of a baby modifies family relationships [18] and changes the behavior and type of care for older children [19], varying from overprotection [20] to the distancing of other children and neglect of parental obligations [21]. PD also affects older siblings who experience feelings of guilt, fear, anxiety and misunderstanding [22]. The couple’s life is also affected as they may become distant from each other, and there is an increase in the frequency of conflicts [23,24]. 

In Western countries, PD grieving is not fully recognized or socially legitimated [25]; it is not granted the importance it deserves, and its consequences are underestimated [26]. Parents do not receive sufficient support on a psychological or spiritual level, and even if death occurs before childbirth [25,27], its invisibility increases the negative repercussions [28]. Within this social context, a father’s grief is even less recognized [29]. On the one hand, the father is imbued with a social role of providing physical and emotional support for mothers, rendering his feelings, experiences and needs invisible [30]; on the other hand, the attachment and bond of a father is considered to be more rational and less passionate [31] since a mother creates her bond with the fetus from the first weeks of pregnancy and this is reaffirmed with fetal movements [32]. However, the risk of dysfunctional grief and the symptoms of psychological trauma is common in both sexes [33,34], although less studied in fathers [29].

The theoretical framework used in this research is based on the Worden theory of the tasks of grieving [35]. Worden defines grieving as an adaptation process in which the person suffering a loss must complete the following “tasks”: accept the reality of the loss, work through the grieving for the loss, adapt to an environment in which the deceased baby is not there and find a lasting connection with the deceased baby while continuing a new life. Worden considers seven determining factors to understand the experience of people suffering from a PD process. Among these factors, great importance is given to social issues and the environment in which the adaptation takes place. Worden included such environmental factors as “mediator 6: social variables” [36]. Although the individual implications of perinatal death on the parents [11,12,13,14,15,16,17,18] and the social support available to survivors [37] have been extensively investigated, there is a lack of research involving fathers [38,39]. Moreover, the impact of PD on the social environment and the extended family has been poorly investigated [40]. The aim of our study is thus to explore, describe and understand the impact of perinatal death on parents’ social and family life.

## 2. Methods

### 2.1. Study Design

A qualitative study was designed based on the hermeneutical phenomenology of Gadamer, which highlights the dialogic nature of understanding. For Gadamer, “understanding” is a process in which the researcher’s viewpoint (“horizon of pre-understanding”) participates in a dialog with the participants’ horizon of pre-understanding [41,42]. As a result, an understanding of the phenomenon arises, and this implies a “fusion of horizons” [41] (p. 305). 

The Gadamerian-based method developed by Valerie Fleming has been followed in this study [43]. Firstly, researchers must ensure that the phenomenon they intend to study can be experienced in the lifeworld. This is the case for perinatal death and its impact on parents’ social and family life. Secondly, the researchers must clarify their pre-understanding of the phenomenon (“reflexivity”). For Gadamer, it is not possible to exclude the researcher’s previous ideas (prejudice or pre-understanding) and all understanding emerges from the researcher’s pre-understanding of a phenomenon. Thus, the researchers’ pre-understanding of the phenomenon has to be clarified and they must explain how such pre-understanding was used in the investigation [44]. Three of the researchers were midwives (M.C.-A., E.H.-S., M.B.C.-F.) who helped us to recruit the sample and carry out the interviews. Other researchers have experience in qualitative research (C.F.-S., J.M.H.-P., J.G.-M., I.M.F.-M.) and have previously researched perinatal death and grief (F.R.J-L., I.M.F.-M.). This helped them to conduct the study and analysis. One researcher is a native Spaniard who is bilingual in English (J.M.H.-P.) and helped us in the process of the translation and back-translation of the interviews to avoid losing their expressiveness.

The study was conducted in the hospitals of Torrevieja and Vinalopó, in Alicante, Spain. Both belong to the Spanish National Health System (public hospitals) and have an average of 1400 births per year each.

### 2.2. Participants

In order to recruit participants, the data manager at both hospitals provided the researchers with the contact information of all the parents who had suffered a perinatal death in the last 5 years (*n* = 63). Once the researchers checked that the potential participants met the inclusion criteria, one of the researchers (M.C.-A.) telephoned them to explain the study’s aim and to invite them to participate. When the potential participants asked for some time to think about the invitation, they were contacted again two days later. The interviewers made an appointment with those who accepted participation. The inclusion criteria were as follows: having suffered a perinatal loss between the 22nd week of gestation and the first week of life of the baby; the death having occurred between three months and five years before the time of the interview; and signing the informed consent form. The exclusion criteria were as follows: not speaking Spanish or English (8 cases) and refusing to participate in the study (18 cases). Nine people declined participation because they did not want to talk about the topic, and another nine said they did not have time to participate. In the end, 21 participants were interviewed (age range 26–43 years old, SD: 4.76 years). The researchers let the participants decide where they wanted to be interviewed so that they felt most comfortable. All in-depth interviews were conducted face-to-face with the participants, who did not have to go back to the hospital where their babies’ death occurred. The participants did not receive any economic compensation for their participation. All of the participants were part of either a nuclear family (*n* = 19) or an extended nuclear family with grandparents (*n* = 2). The socio-demographic data of all participants can be seen in Table 1.

### 2.3. Data Collection

In-depth interviews were carried out by three researchers working as midwives (M.C.-A., E.H.-S., M.B.C.-F.) in the hospitals in which the study was conducted. The researchers received training on how to make the interviews resemble an in-depth conversation and on how to handle painful situations. They had an interview protocol that contained the aims, ethical issues, and a list of questions (Table 2). Before commencing with the interviews, the researchers reminded the participants of the aim of the study and the voluntary nature of their participation. All participants signed an informed consent. All interviews were recorded for later transcription. The average duration of the interviews was 45 min. The researchers made notes on nonverbal elements of the communication. When the researchers considered that data saturation had been reached, the data collection was finalized. This decision was made upon the following two criteria: 1) there were no new issues arising from the participants’ interviews, and 2) the remaining eligible participants did not have different sociodemographic characteristics (for example, single parents). Those who did not participate despite being eligible and giving consent to participate were sent an email to thank them for their willingness and to let them know that it was not going to be necessary to interview them.

### 2.4. Data Analysis

The following steps described by Valerie Fleming [43] were followed in order to understand the phenomena from the data. Firstly, the researchers (M.C.-A., E.H.-S., M.B.C.-F.) reached a spontaneous understanding through the dialog with the participants (interviews). In order to do so, the researchers made notes that were useful in the subsequent transcription and coding process. These notes also helped to modify some questions or request more details which were not initially included in the interview protocol. Secondly, the researchers gained an understanding of the studied phenomenon through the transcription analysis. All interviews were transcribed (M.C.-A., E.H.-S., M.B.C.-F.) and incorporated into a Project in ATLAS.ti 8 Software for analysis (C.F.-S., J.M.H.-P., F.R.J.-L., M.B.C.-F.). The transcripts were read to get a general idea of what the participants said. Then, the transcriptions were re-read line by line for detailed analysis. Significant fragments were selected as citations. The citations were coded using the ATLAS.ti coding procedures (open coding, coding from a code list, autocoding). The initial codes were grouped into units of meaning, sub-themes and themes using ATLAS.ti functions such as grouping codes or creating networks. An example of the coding work that led us from the data (quotation) to the subject can be seen in Table 3. Thirdly, to increase the rigor of the study (credibility, reliability), some participants checked the units of meaning, themes, and sub-themes.

### 2.5. Ethical Considerations

The study was approved by the Joint Investigation Commission of the Hospitals of Torrevieja and Vinalopó (CIVHTV-16-28-12). Participation was voluntary. Following the interview protocol (see Table 2), information on ethical aspects (i.e., confidentiality, the need to record interviews, the possibility of not responding or abandoning the interview, anonymity) was provided at the beginning of the interview. The participants received emotional support during and after the interviews.

## 3. Results

The data analysis led to 36 units of meaning that were grouped into seven sub-themes. These sub-themes were grouped into two main themes that helped us to understand how parents who have suffered a PD perceive the impact it has on their social and family environment (Table 4).

### 3.1. Perinatal Death Affects Family Dynamics

In the context of the nuclear family, parents and siblings who live in the same house suffer the consequences of perinatal death individually. All suffer a grieving process after the loss of the baby. However, there is also an impact of perinatal death on family dynamics as it is impossible to separate them from the emotional, physical and psychological state of each of its members. This theme represents the participants’ perceptions of how family dynamics have been affected. It includes the role of the father, the siblings and the way the couple’s ties have altered.


*“(…) Late at night our looks crossed, and he said: ‘Darling I’ve also been remembering’. And I: ‘Yes, I know you’ve been remembering’. But he doesn’t want to show that something is hurting”.*
(P-10)

#### 3.1.1. The Father: The Struggle between Conserving the Stereotypical Protective Role and Succumbing to Pain

Participants perceive the influence of social stereotypes in Spain on the distribution of roles traditionally given to fathers and mothers within a family. According to these stereotypes, the responsibility for the protection of the family rests on the father, and in the case of a painful situation such as PD, they are responsible for caring for the mother. Traditionally characterized as strong, tough and less sensitive, many fathers do not publicly show their feelings. Furthermore, some women think that men go through a different grieving process as they have not gestated the baby and believe that the bond is less strong than that of women.


*“I think that, even if it affects them… the feeling is different. You *[the woman]* have experienced it for a long time, that bond that you have created. As it wasn’t born, they have not felt it and it is different, you have felt it and the man only puts his hand there”.*
(P-14, Female)


*“At least try to be the stronger, because I thought that, ‘Poof! How can I feel as down as she does…’ Maybe because I am the man (we have) to swallow it and stand on our feet or be a little stronger or (…) try to cry apart”.*
(P-4, Male)

However, the implicit demand for fathers to be strong raises tensions because men also go through a grieving process with very deep emotions, thoughts and feelings. Despite trying to hide or disguise these feelings, everyday life makes it impossible to disguise reality—a reality that reflects a suffering similar to that of mothers, and a pain that is difficult to alleviate, to which the fathers have to add the burden of having to carry both their “weight” and that of their partner. Participants describe the process by which the father ends up succumbing to pain as “falling apart”:


*“He tried to make himself strong, although occasionally I found him crying in the corner, where he thought I would not see him (…)”*
(P-16)


*“In the ten years that I’ve been with him I’ve never seen him cry... he tried to become strong and brave, but he couldn’t, and he fell apart”.*
(P-14)

The father participants also perceived the pressure of having to be stronger and ended up rebelling against this stereotypical requirement. Some no longer showed a frustrated attempt to be stronger or an inability to maintain appearances. They simply rebel against the stereotype of a strong and insensitive father:


*“But supposedly fathers have to be stronger than them. Stronger than them? Shit! That’s not true. We fathers have our feelings too, you know?”*
(P-7)

The father participants noted that there is a lack of support and a certain abandonment of the task of looking after them. During hospitalization and immediately after PD, both physical and psychological care was directed towards the mothers. Although one father reported that he felt he was very well treated, normally they faced an absence or lack of care. This neglect hindered the grieving process and made the father’s needs invisible while engulfed in the strong emotional pain caused by the experience of losing and “burying” a baby. The invisibility of the fathers’ grief makes it more difficult for them to work on the pain they feel after the loss.


*“Nobody cared about my husband. Because I was bad and he was more worried about me. So my husband couldn’t lean on anyone.”*
(P-19)


*“She was given more attention, when it came to explaining things, when it came to telling things. Most of the time it was more towards her, (…) I was invisible to many people. However, I was also having a hard time.”*
(P-20)

#### 3.1.2. The Elder Siblings: Feeling Overprotected or Abandoned

Together with the parents, the elder children in the family are also generally forgotten by the care system. In general, they do not receive any kind of support and suffer the consequences of the grief that their parents are going through; nobody pays attention to them, and nobody notices whether they need support in order to understand that they have lost a brother or sister. Some parents’ stories show excessive protection towards their other children for fear of something happening to them.


*“We are more protective and spoil him a little more. For example, before he had to do a job to get something. But not now. Now he can always easily get something out of me…”*
(P-18)


*“I already know what it is to lose a child, I have to take more care of them. My daughter tells me: ‘Why do you take so much care of me now?’, and I tell her I’m afraid that something could happen to her or someone could do something to her; it makes me very scared.”*
(P-10)

However, other testimonies are in stark contrast to the idea of overprotection: these statements show parents who have partly neglected their older children because they were so immersed in their own grief that they gave up their duties and responsibilities as parents of other children. The pain, suffering and depression which a mother and a father who lose a baby experience can afflict them not only physically and psychologically but also behaviorally, leading to the neglect of other children.


*“My children told me, ‘Mommy, you’ve changed a lot, you’ve become indifferent to us, (...) I understand your pain and everything, but you’ve put us aside, you only cared about the little girl, and we need you’. And all that was true.”*
(P-2)


*“… [My children] sometimes dared not talk to me because I didn’t feel like doing anything. I apologized to them because I turned away from them and became distant from them. I gave in so much to my pain, and this was the way of living my grief.”*
(P-12)

In most cases, parents failed to inform their children of what had happened. They omitted information and avoided facing the difficult moment of telling their children what had really happened. They told them nothing in order to protect themselves, to avoid reviving or rekindling the suffering, and to protect their children from this suffering and prevent them from feeling pain; on the other hand, some parents did not explain the situation because they thought the children were not ready to understand what had happened, postponing the explanations to when their children were older.


*“He had a hard time telling our daughter. I remember being there, he told me that ‘Now how do I explain it to you, how can I tell you?’ But do you know what’s happened? that she hasn’t seen the baby (...) I haven’t shown her a photo. We chose not to say anything, why make her suffer?”*
(P-19)


*“I didn’t say anything, because I thought she was strong, it was hard for me to tell my daughter things I’m not sure she’ll be able to understand. She asked me what was wrong with me and I told her that when I was older I would tell them.”*
(P-10)

In the cases in which the children understood what had happened and had been told about it, they went through a grieving process, which, according to the parents, was very hard for them as parents as they could see their children were suffering. Tears, anxiety, and fear were the common reactions of most of the children, who had been looking forward to having a baby brother or sister. One father also said that the brother was trying to keep the dead baby in mind, by remembering, talking about him, representing him as an angel, or drawing him.


*“He always remembered and said: “Mom, and my little brother who is now in heaven? (crying) (...) and then he reminded me of him, and I tried not to cry, but I saw him so small that we both ended up crying.”*
(P-6)


*“My son had a hard time. He got rather depressed. He still says his name today and makes drawings of his brother with (…), with his wings and such things.”*
(P-3)

In one case, a participant reported self-protective reactions by the siblings, such as thinking about avoiding a bond with other living siblings in order to prevent suffering in case of a new loss.


*“It affected him a lot, and he told his little brother, who was in my belly, that he was not going to love him like he had the other one. I think he didn’t want to become fond of him in case it happened again.”*
(P-5)

#### 3.1.3. New Pregnancies Dominated by Fear: Medicalization and Avoidance

Families who tried to have another baby again after losing a child faced a serious challenge. The new pregnancy brought on feelings of fear, stress and anxiety regarding the possibility of losing another baby. The thoughts and feelings of the previous experience emerged and rekindled fears, and instead of being something happy and wonderful for the family, the new pregnancy became something that overwhelmed them. Behaviors such as the medicalization of the pregnancy appeared, with continuous pregnancy controls and excessive abuse of emergency services out of fear of another loss.


*“… Even if they told you that everything is going well, every month each time I went to the ultrasound, well, imagine. I was outside *[in the waiting room]* and my legs were shaking.”*
(P-15)


*“I went through the next three pregnancies in fear. A pain, *[and I went]* to the emergency room; a spot, to the emergency room (...). I went very often, I was very afraid. I was in panic in case I went through the same thing again.”*
(P-16)

Other participants reported that the fear of a repeated painful experience led them to avoid another pregnancy. Although they recognized that they would like to have another baby or that it could be positive to experience another pregnancy, the fear of going through the same PD situation was strong, and they even avoided thinking about it.


*“I would like to have a baby sometime, but I don’t want to go through the same thing again. I especially don’t want my wife to go through it again. I don’t know whether she could take it. So I avoid these thoughts.”*
(P-18)


*“It may be positive for me to get pregnant again, but on the other hand I am very scared of going through that situation again. I don’t want to go through that situation again, it distresses me to think about it.”*
(P-19)

#### 3.1.4. The Strengthening and Weakening of the Couple’s Bond

This difficult experience was able to strengthen the bonds of the majority of the participants. Some reported what happened as something that they had to live through together. Others, perceiving the risk to the couple, consciously decided to strengthen their bond; i.e., to stick together. Thus, athepainful experience ended up strengthening the bond of the couples who decided to face it together. This was recounted by two participants: 


*“We’ve talked a lot, and I’ve told him: ‘What has happened to us is something that we’ve been through together’. I have the feeling that it has brought us much closer.”*
(P-20, Male)


*“We clung more to each other and became stronger. (...) We stuck together, *[we think:]* ‘do they want to hurt us? Well, nobody will be able to.’ Because I know this can break a marriage or a family.”*
(P-16, Female)

However, in the first months after the PD, arguments and disagreements became more frequent. This seems to be because, according to the participants, each member of the couple faces the pain and stress of the PD in their own way. Pain, together with almost continuous emotional stress and tension, triggers quarrels and disagreements.


*“We have more arguments than before, many more. Every time when I lose my temper because I am a little low or depressed or down, we argue.”*
(P-4, Male)


*“... I have my own character, and we didn’t see it in the same way. And he shouted and said to me: Stop, don’t carry on! And interrupted me. (...) And I was enraged because I get angry when people shout at me.”*
(P-19, Female)

The couple’s sexual relations also changed. During the grieving process, a lack of sexual desire was a common point in almost all of those interviewed. This, coupled with the fear of a new pregnancy and fear of going through the same thing again, made sexual intercourse null or minimal, although over time this improved.


*“Well... we were doing nothing for a while because I didn’t feel like it, how can you feel like it? You have your head somewhere else... when you’re sad, who is going to feel like it?”*
(P-1)


*“At first I didn’t want to have sex, I didn’t feel like it, and I thought, what if I got pregnant? I didn’t feel like anything, I was afraid to go through it again, so I avoided sleeping with him.”*
(P-6)

### 3.2. The Social Environment of the Parents is Severely Affected After Perinatal Death

This section covers three sub-themes that reflect how PD affects the parents’ social sphere. The impact of the loss reaches the members of the extended family as well as work and social environments. Both seeing other children and the use of stereotyped messages of comfort intensify the pain. In addition, stereotyped messages are not effective because they denote an intrinsic banalization of the loss or the disallowance of grief. All this leads to parents avoiding social contact.


*“We can’t do anything because there are babies everywhere, pregnant women everywhere… And we get depressed. So you don’t know where to go, and you lock yourself in the house.”*
(P-17)

#### 3.2.1. Impact of the Perinatal Death on the Extended Family

The extended family, understood as the network of relatives—especially blood relatives—which extends beyond the domestic group, suffers the impact of the pain of their relatives and the death of a baby they did not get to know. Participants report receiving condolence and support from their relatives and that they perceived double pain for the mother/father and for the deceased baby:


*“The family called us, and they worried about us, my sister, my aunt (pause) saw that she had hit rock bottom, and that’s why they called (...). I also imagine that they would be feeling bad. It was their nephew, and suddenly they don’t have him. They would call me and them (...) I know they had a bad time.”*
(P-7)


*“I had never seen my mother (paternal grandmother) so bad, so bad (…).”*
(P-9)

One participant recounted how her mother (the grandmother of the deceased fetus) developed her own grief for a similar loss decades before when she had no opportunity to experience grief. Losing her grandchild enabled her to remember her loss and resolve a pending grief.


*“My mother lost a daughter and had no chance to grieve. She has grieved for her own daughter with her granddaughter. She had lost a daughter who had been stillborn thirty-five years before, and they didn’t want to teach her to grieve. They buried the baby immediately. So it’s been good for her, to mourn now.”*
(P-13)

#### 3.2.2. Impact on Work: Lose a Child, Lose a Job

Undoubtedly, the work environment of people suffering from PD is damaged. People in their grieving process reported a decrease in their performance at work, causing, in two cases, the loss of employment. In addition to losing a child, they had to face job loss and its consequences, increasing their pain, anguish, stress and suffering. The pain experienced by the people who lost a baby lowered their ability to concentrate, and this, together with the deprivation of the quantity and quality of sleep, explained how their performance worsened.


*“It gave me bad depression, and I lost my job. I lost my job because they had to give me more days off, and they told me that they couldn’t wait any longer, and so I lost my job (...) everything combined, my baby died, and they fired me... It was one problem after another.”*
(P-2)

For one father, leaving work was the result of a choice between staying with his partner or going to work in an unsuitable physical condition, suffering from insomnia or a lack of concentration.


*“I lost the habit of sleeping, I had to choose, let’s say I had to quit my job to be with my wife. (...) my wife wanted me to start... that I went to work, right? (...) I was going to work without sleep day after day and working, so think of it, working without sleep and with my head in another place.”*
(P-11)

Other participants presented their work as a means of escape or isolation from other environments such as parks or family gatherings, where the elements that recall the lost child are present.


*“You are looking for your escape routes, either working (…) You isolate yourself a little from all of that because what you want is to avoid environments that remind you of your child.”*
(P-5)

#### 3.2.3. Social Impact: Between Disenfranchised Grief and the Remembered Pain

Parents who have suffered the death of their babies return to their social environment, beyond the family and work environment, where they must complete the tasks of working through their grief and adapting to life without the lost baby. In this environment, they may find that the attitudes and responses of their acquaintances to their pain do not recognize or banalize PD. Thus, the pain of the parents becomes invisible and is minimized with comments including that they had not got to know their dead baby or that he or she was very small.


*“Then people tell you: ‘it’s better that it happened when the baby was three days old and not three months’, can you believe it? As if it hurt more before or after, you know? People are incredible.”*
(P-7)

Participants perceive that it is not socially recognized that the bond with babies begins from the beginning of pregnancy, rather that this relationship is related to the birth and presence of the newborn baby in the family. Parents experience grief with suffering and are also disenfranchised by the social environment to which they belong, which makes it difficult for them to express their grief naturally. Underestimating grief could hamper fathers’ and mothers’ ability to take actions toward a healthy grief (for example, accepting reality).


*“When I talk with friends, family, I try to make the effort, (…), to talk about my son and that I have been a father. Although nobody recognizes this, I am sure, I have felt responsible as a father, I have had the feeling of being a father.”*
(P-20)

It can be painful and strange for mothers and fathers that their acquaintances may behave as if nothing had happened; this is not due to a need for the validation of pain but rather because they interpret the absence of mention of the loss as not recognizing it. A mother recounted the following:


*“I don’t need anyone to validate my grief. But people frequently behave as if nothing had happened to you. People say that ‘as it was a baby you didn’t know... you know? You didn’t want it’ (pause) and I said: ‘Let’s see, didn’t you see my belly? (…) You can’t behave as if, as if nothing had happened.”*
(P-13)

This non-recognition by the community increases with phrases, comments and actions that, far from helping parents, cause them increased pain and suffering. They hear hurtful phrases that downplay the death of their baby, increasing or blocking the pain they suffer, and unpleasant comments that undermine the loss by downplaying it or considering that their painful experience is not unique, or suggesting that another baby will dissipate the pain for the lost baby.


*“I was told: ‘Ah, don’t worry, in a year or two you will have another one’. ‘The same thing happened to my cousin, to my neighbor…’ I don’t care what happened to your cousin or your neighbor! I care about myself, and what has happened to me hurts.”*
(P-8)

Perhaps because of this, the majority of the families in the process of grieving preferred to live alone, isolated from a community which sometimes failed to understand their pain. Some participants reported that they isolated themselves for a while, avoiding going out and stayed at home or with their closest relatives more—they tried to work through the experience alone.


*“It makes you shut yourself up a little, that is, go out less, feel like being alone, withdraw a little (...). It has also isolated us from friends (...). My partner and I shut ourselves up at home more, inside ourselves.”*
(P-4)

Other participants reported that they suffered stress by exposing themselves to social situations that reminded them of their deceased babies. Birthdays, baptisms, walks in the park or meeting pregnant mothers were situations in which it was common to see children of all ages including babies. These situations reminded them of their dead baby and were a source of stress for parents. They were difficult situations that they tried to avoid in an attempt not to increase their suffering.


*“The horrible birthdays, *(sob)* but the worst thing is Christmas, when we get together with my nephews and nieces. (...). Because you see the children and say: ‘Oh no! Our baby could have been here’”*
(P-12)


*“Wherever you go... you are reminded, and the child is not there, not there *[silence]*. It’s not there, and it’s frustrating (...). So we avoided being in places where there were children. It brought us the anxiety of remembering again... We had lost ours two months before”*
(P-5)

## 4. Discussion

The aim of this study was to describe and understand the impact of perinatal death on parents’ social and family life. Using Worden’s theoretical framework [35] has allowed us to approach environmental factors (i.e., social and familiar) that play an important role in the process of adaptation of those who have suffered a loss due to PD. However, our study suggests that the environment is not only the physical space to which people adapt; the parents’ environment also suffers the impact of the PD and contains elements that can facilitate or obstruct the adaptation process. The results of this study suggest the need to reinterpret the role of the father, which is traditionally relegated by society and professionals to a secondary role in relation to the children [28]. In line with other studies [45,46], we can see that fathers perceive that their pain is undervalued. However, in this study, mothers also openly acknowledge that fathers fall apart and give up their protective role in order to work through their grief. In addition, men—culturally recognized as the pillar of support for mothers and the family [47]—reported that they needed someone who they could ask about their condition or get emotional support from because, as shown in other studies, they noticed that attention was centered on the mother [47,48]. This may make some fathers feel marginalized from the grieving process [11] and hinder their socialization, which is needed for a successful grief process [35]. This invisibility of the father’s grief may be due to the fact that, when pregnant, the woman has a physical connection to the baby while the father’s link is only mental [49,50]. Health professionals and the community ignore and disallow the father´s grieving for PD although it has been shown that PD does have repercussions for fathers [46]. It is necessary to determine the needs of fathers [51,52] and clearly recognize their grief [38,39,47,53]. Although the sample of fathers in this study as more limited than that of mothers, our study investigates the experiences and perceptions of both sexes.

PD also impacts siblings, who are sometimes omitted or given incorrect information about what has happened [54]. In this study, the parents reported that they chose to omit information out of a sense of protection, although in other cases they decided to wait for the children to mature so that they could understand what had happened. However, it is necessary that minors can also grieve for the death of their siblings [55], and this starts from recognizing the loss instead of denying it [35]. Healthcare providers must take into account the siblings’ needs and feelings [56]. When perinatal death occurs, parents change their way of caring, and this affects their children [57]. Some parents in this study reported that they were absent and neglected their older children and felt guilty about it. As has been previously reported, other parents reported that they overprotected their children out of fear of something happening to them [20,21,58,59].

PD has repercussions in subsequent pregnancies [60], which is a complex situation that requires health professionals’ support [61]. In some studies, mothers and fathers expressed that a new pregnancy would decrease their pain and emotional emptiness [19,62]. However, our study participants fear that a new pregnancy would have a similar outcome and this causes some parents to dismiss that possibility. In this regard, a new pregnancy without the mother being ready can cause attachment problems with the next child [25,63]. Participants who did become pregnant after the PD reported that they went to emergency services more often in the following pregnancies, out of fear and concern that the PD would recur. Other studies suggest that fear and anguish of a new PD increases visits to healthcare services [15,64,65].

PD has an important impact on the dynamics of the couple’s relationship [39,66]. Previous studies show that abortion or PD situations increase disputes between couples [67] and increases the probability of a divorce [68], especially when one of the couple does not feel the necessary support from their partner [23]. However, this study suggests that the PD served as a stimulus to strengthen the bond of the couple (face adversity together) but that it also increased the number of arguments between the couple. In accordance with other studies in which the sexual life of couples was shown to change after PD or abortion [69], sexual desire and sexual intercourse decreased according to the participants of this study.

Perinatal death and grief not only affect the nuclear family but also the extended family and the parents’ social environment [70]. Grandparents’ expectations and hopes for the arrival of a new grandchild are jeopardized, which could have consequences that need to be addressed and understood [71,72]. Grandparents and close family members play a crucial role in social support and communication with parents [73]. Perinatal grief in grandparents and close relatives is a field to be investigated since this could help mitigate the consequences of PD. Therefore, the design of strategies to support family members is crucial [74].

The impact of PD extends to the workplace; a study [75] conducted in the UK calculated the losses linked to absenteeism and the decrease in productivity. In contrast, in this study, participants reported their own decisions to leave their job as well as cases of being dismissed due to increased absenteeism. However, an increase in dedication when using work as a refuge was also shown. The community (social context) of the parents perceives the PD in a totally different way from the parents themselves [76]. The recognition of grief is often avoided and the pain of the parents is silenced, resulting in a sense of unauthorized grieving [62,77]. This taboo and undervaluation of grief was perceived by the mothers and fathers in the study when they received comments that trivialized their pain. This made this type of grief invisible, increasing the sensation of stigma [78,79]. Some families reported that there were people who shunned them and avoided talking about the subject, since this grief is different from other types of loss: no flowers are given, no cards are offered, there are no visits and there are no religious rituals to validate the grief [25]. However, making this problem visible and raising awareness in society may be important for parents [80,81].

The main limitation of this study is that parents were interviewed while they were grieving for PD, which includes fetal and postnatal death—two different situations that are likely to trigger different responses. However, it was preferred that the inclusion criteria were broad (i.e., PD in general); furthermore, as few parents suffer postnatal death, creating a study solely including such parents and comparing discourses would be difficult. An additional limitation was that the sample in this study was mostly made up of parents of Spanish nationality; including other cultural backgrounds would have extended the generalizability of the data. Although both fathers and mothers were interviewed, as suggested by other studies [51,52], the sample of fathers was smaller than that of mothers as some refused to discuss the issue or could not because of work issues. Parents who declined participation could have provided valuable information that in some cases could have been associated with a pathological grieving process. The study reflects the parents’ perceptions of the impact of their child’s PD on their social environment. This impact could have included other social actors such as siblings, grandparents, etc. However, ethical issues (interviewing children) and accessibility (grandparents, obtaining parental permits) forced us to contemplate these perspectives through the parents’ perceptions. All participants were part of either a nuclear or an extended nuclear family (with grandparents)—none of the participants were single parents or couples in non-heterosexual relationships. Interviewing participants from single-parent families (or any other type of non-normative families) could have provided complementary results.

## 5. Conclusions

PD affects family dynamics, including a couple’s relationship, which can be strengthened by coping with adversity together or weakened by increased arguments or decreased sexual activity. Siblings and grandparents also suffer the consequences of PD in the family. Feelings of abandonment or overprotection may arise in older siblings and expressions of grief or even an opportunity to resolve past grief in grandparents.

Regarding their social environment, parents who suffer a loss due to PD experience work problems, a lack of recognition of their grief, the absence of rituals, the trivialization of their loss and the disavowal or delegitimization of their mourning.

In terms of the practical implications of this study, a family that suffers a PD may need comprehensive care that goes beyond physical or psychological care. Knowing the impact of PD on family dynamics, as well as the social, family and work environment can be useful for social care providers, health care professionals (as nursing, midwifery or medicine) and educators. Attention must be paid not only to mothers but also to fathers, older siblings and the extended family. Policies that give the deceased fetus or baby the right to be registered and to have an identity or name should be developed. Religious rites should be offered in hospitals and communities for those parents who wish to celebrate them. At a community-care level, nurses, social workers, occupational therapists or psychologists should offer targeted interventions to foster a healthy grief process amongst families who have suffered a perinatal loss. The recognition of the loss by the community would help the families to work through their grief.

## Figures and Tables

**Table 1 ijerph-17-03421-t001:** Socio-demographic data of the participants (*N* = 21).

Participant	Age	Sex	Nationality	Employment	Elder Children (Age Range)	Previous Loss (n)	Moment of Death	Age of Baby
P-1	26	Female	Colombian	Employed	No	No	Intra-natal ^1^	40 weeks
P-2	43	Female	Colombian	Unemployed	Yes (8)	No	Ante-natal ^2^	30 weeks
P-3	38	Female	Spanish	Employed	Yes (8)	Yes (2)	Intra-natal	24 weeks
P-4	38	Male	Spanish	Employed	No	Yes (2)	Intra-natal	24 weeks
P-5	37	Female	Spanish	Employed	Yes (5)	No	Ante-natal	34 weeks
P-6	43	Female	Spanish	Employed	Yes (6–12)	No	Post-natal ^3^	6 days
P-7	43	Male	Spanish	Employed	No	No	Post-natal	6 days
P-8	33	Female	Spanish	Employed	No	No	Post-natal	3 days
P-9	33	Male	Spanish	Employed	No	No	Post-natal	3 days
P-10	31	Female	Ecuadorian	Unemployed	Yes (5)	Yes (1)	Ante-natal	28 weeks
P-11	26	Male	Spanish	Employed	No	No	Ante-natal	40 weeks
P-12	30	Female	Spanish	Unemployed	Yes (3)	No	Ante-natal	40 weeks
P-13	36	Female	Spanish	Employed	No	No	Intra-natal	24 weeks
P-14	33	Female	Spanish	Employed	No	No	Ante-natal	36 weeks
P-15	37	Female	Spanish	Employed	No	Yes (1)	Ante-natal	34 weeks
P-16	37	Female	Spanish	Unemployed	Yes (6–12)	No	Ante-natal	38 weeks
P-17	38	Male	Spanish	Employed	Yes (6–12)	Yes (1)	Ante-natal	38 weeks
P-18	38	Male	Spanish	Employed	Yes (7)	Yes (1)	Ante-natal	37 weeks
P-19	35	Female	Spanish	Employed	Yes (7)	No	Ante-natal	37 weeks
P-20	35	Male	Spanish	Employed	No	No	Ante-natal	38 weeks
P-21	37	Male	Spanish	Employed	No	No	Ante-natal	25 weeks

^1^ Intranatal death: fetus death during birth. ^2^ Antenatal death: fetus death between 22 weeks of gestation and birth. ^3^ Postnatal death: baby’s death between birth and the first week of life.

**Table 2 ijerph-17-03421-t002:** Interview protocol.

Stage	Subject	Content/Example Questions
Introduction	Motives, reasons	Knowing their experiences to be able to help other parents in their situation.
	Ethical issues	Inform about volunteering, recording, consent, possibility of dropping out.
Beginning	Introductory question	Tell me your experience. What happened to you?
Development	Conversation guide	How has your baby’s death affected your family? How has it affected your circle of friends? How has it affected your environment?
Closing	Final question	Is there anything else you’d like to tell me?
	Appreciation	Thank them for taking part. Remind them of how their testimony will be used and tell them we are at their disposition.

**Table 3 ijerph-17-03421-t003:** Example of the codification process.

Quote	Initial Codes	Unit of Meaning	Subtheme	Theme
*At the beginning he said, ‘No mommy, nothing’s going to happen, everything is going to work out well, we’re going to get over it, but... there came a time when he fell apart* (P-2)	Father: encourage, Father: protect, Father: fall apart	Father: Role of protector	The father: struggle between conserving the stereotypical protective role and succumbing to the pain	Perinatal death shakes up the family dynamics
*I already know what it is to lose a child, I have to take care of them more. My daughter tells me: ‘Why do you take care of me so much now?’ I tell her that I’m scared something will happen to her or someone will do something to her. It scares me a lot* (P-10)	Other children: take greater care of them. Other children: fear of something happening to them. Other children: notice over-protection	Siblings: over-protection	Elder siblings: between over-protection and abandonment	

**Table 4 ijerph-17-03421-t004:** Themes, subthemes and units of meaning.

Main Theme	Subtheme	Units of Meaning
Perinatal death affects family dynamics	The father: struggle between conserving the stereotypical protective role and succumbing to pain.	Masculine figure. Feelings. Father: lack of support. Father: traditional role. Father: protective role. Support. Father: forgetting.
	Elder siblings: feeling over-protected or abandoned	Siblings: overprotection. Abandonment of children. Affecting siblings. Siblings: concealment. Brothers: self-protection. Siblings: memories. Siblings: making a drawing.
	New pregnancies dominated by fear: medicalization and avoidance	Fear of new pregnancy. Avoiding new pregnancy. Medicalization. New pregnancy. Fear. Panic. Repressing desires.
	Between strengthening and weakening of the link between the couple.	Marriage. Increase in quarrels. Alteration of sexual relations. Strengthening of bond.
The social environment of the parents is severely affected after perinatal death	Impact of perinatal death on the extended family.	Affecting family. Grief of grandparents. Weeping. Solidarity. Seeing their own children suffering.
	Work impact: lose a child, lose a job.	Loss of job. Reduction of remuneration.
	Social impact: between disallowing grief and remembering pain.	Unrecognized grief. Inappropriate social messages. Social isolation. Avoiding the presence of children.

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
