# Peer review of "Impact of Perinatal Death on the Social and Family Context of the Parents"

_ijerph, 2020, doi:10.3390/ijerph17103421_

Round 1
Reviewer 1 Report
This is an important and interesting study which should prove to be of interest to the readership of this journal. I do, however, have some queries which largely focus on the process of recruitment to the study:
- There was a 29% refusal rate - given the topic area, this is not surprising, but do the research team have any demographic or other information about those who decided not to participate? If so, it would be important to review the findings in light of these, with a particular focus on any limitations. It could be that those who chose not to participate reflected those with a particularly complicated grief response.
- There is little information provided in the paper about the process for recruitment of the participants. How were participants originally contacted? What were the opportunities provided for them to consider their participation? What was the process taken to gather informed consent?
- 21 participants were invited to interview. Given the total participant sample size and accounting for those who didn't meet eligibility criteria or who decided not to participate - this suggests that 16 participants indicated that they wanted to participate but they were not eventually invited (due to an assessment of saturation). How were these volunteers notified that they would not be invited to participate?
- Given the findings about the implications for a nuclear family - more information is needed about the participants in the study with respect to their demographic information. How many of the participants were part of a 'nuclear family'? Were there any single parents included? It is noted where there were older children in the family - but their ages are not included. Presumably there could be different grief responses for older children when compared to younger children? Had any of the participants experienced previous losses? Bearing this in mind, do the team consider the data can be described as saturated across all the themes?
- A definition of 'intranatal', 'antenatal' and 'postnatal' is needed for those unfamiliar with the field.
- There is a theoretical underpinning provided at the onset of the article (Worden) - but this is not referred to in the discussion and is not utilised as a lens through which to further interpret the findings. As an inductive process of analysis was undertaken, did the Worden model prove unhelpful? If not, how did the model help the research team to understand the findings?
Author Response
This is an important and interesting study which should prove to be of interest to the readership of this journal. I do, however, have some queries which largely focus on the process of recruitment to the study:
Response: Thank you for your constructive feedback, which we strongly believe has helped to improve our paper towards a potential publication. All your comments have been addressed and the changes made in the manuscript have been marked in red.
There was a 29% refusal rate - given the topic area, this is not surprising, but do the research team have any demographic or other information about those who decided not to participate? If so, it would be important to review the findings in light of these, with a particular focus on any limitations. It could be that those who chose not to participate reflected those with a particularly complicated grief response.
Response: We have provided additional information about the 18 cases who declined participation in our study. We cannot provide their sociodemographic data because they are not participants in our study and we do not have permission to publish that information. However, we do provide the reason as to why they decided not to participate. Furthermore, we included the following statement in the limitations: “Not being able to interview parents who declined participation could have provided valuable information that in some cases could have been associated with a pathological grieving process”.
There is little information provided in the paper about the process for recruitment of the participants. How were participants originally contacted?
Response: We have described the recruitment process in more detail. Please see section 2.2 Methods:
In order to recruit participants, the data manager at both hospitals provided the researchers with the contact information of all the parents who had suffered a perinatal death in the last 5 years (n=63). Once the researchers checked that the potential participants met the inclusion criteria, one of the researchers (MC-A) telephoned them to explain the study’s aim and to invite them to participate. When the potential participants asked for some time to think about the invitation, they were contacted again two days later. The interviewers made an appointment with those who accepted participation.
What were the opportunities provided for them to consider their participation?
Response: We have provided more information about how the interviews were conducted and the options we gave the participants. Please see section 2.2 in the manuscript.
What was the process taken to gather informed consent?
Response: We have included information about the process taken to gather informed consent in the data collection section of the manuscript (please, see section 2.3): “Before commencing with the interviews, the researchers reminded the participants of the aim of the study and the voluntary nature of their participation. All participants signed an inform consent”.
Additionally, we inform about ethical aspects of participants’ participation in the section “ethical considerations” (section 2.5) of the manuscript: Following the interview’s protocol (see Table 2), information on ethical aspects (i.e. confidentiality, the need to record interviews, the possibility of not responding or abandoning the interview, anonymity) was provided at the beginning of the interview.
21 participants were invited to interview. Given the total participant sample size and accounting for those who didn't meet eligibility criteria or who decided not to participate - this suggests that 16 participants indicated that they wanted to participate but they were not eventually invited (due to an assessment of saturation). How were these volunteers notified that they would not be invited to participate?
Response: We have clarified this in the data collection section (section 2.3) of the manuscript: “Those who did not participate despite being eligible and giving consent to participate were sent an email to thank them for their willingness and to let them know it was not going to be necessary to interview them”.
Given the findings about the implications for a nuclear family - more information is needed about the participants in the study with respect to their demographic information. How many of the participants were part of a 'nuclear family'? Were there any single parents included? It is noted where there were older children in the family - but their ages are not included. Presumably there could be different grief responses for older children when compared to younger children? Had any of the participants experienced previous losses? Bearing this in mind, do the team consider the data can be described as saturated across all the themes?
Response: All these comments have been addressed in different sections of the manuscript:
- We have added the following statement: All of the participants were part of either a nuclear family (n=19) or an extended nuclear family with grandparents (n=2).
- We have added a table to the column under the heading “Previous perinatal losses”.
- We have added data into an existing column (for example: sibling’s age or sibilings’ age range)
- We have added the following limitation of our study: “All participants were part of either a nuclear or an extended nuclear family (with grandparents). None of the participants were neither single parents, nor couples in non-heterosexual relationships. Interviewing participants from single-parent families (or any other type of non-normative families) could have provided complementary results”.
A definition of 'intranatal', 'antenatal' and 'postnatal' is needed for those unfamiliar with the field.
Response: We have provided these definitions at the foot of table 1: 1 Intranatal death: foetus death during birth. 2 Antenatal death: foetus death between 22 weeks of gestation and birth. 3 Postnatal death: baby’s death between birth and the first week of life.
There is a theoretical underpinning provided at the onset of the article (Worden) - but this is not referred to in the discussion and is not utilised as a lens through which to further interpret the findings. As an inductive process of analysis was undertaken, did the Worden model prove unhelpful? If not, how did the model help the research team to understand the findings?
Response: In the results section, we make reference to the concepts of Worden’s theoretical framework that are of importance in our work in order to highlight the inductive nature of our research (for example: grief tasks, grief elaboration). However, we have not cited the author because this goes against standard procedure in academic writing.

Reviewer 2 Report
2.2 Participants
16 of the eligible cases are not accounted for in the description
Tables 2 and 3
Difficult to discern how the columns relate to one another due to spacing
Author Response
Thank you for your constructive feedback, which we strongly believe has helped to improve our paper towards a potential publication. All your comments have been addressed.
2.2 Participants.
16 of the eligible cases are not accounted for in the description
Response: Information on the non-eligible individuals has been provided. Sociodemographic data is not included in the table as these refer to the “participants” who finally participated in the study. No data is provided as they were ultimately not study participants. However, it was verified that there were no demographic characteristics that could provide different or interesting data (for example, single parents, etc.). As the sociodemographic characteristics were similar, their participation was discarded when the saturation of the data was reached.
Tables 2 and 3
Difficult to discern how the columns relate to one another due to spacing
Response: It is true, being tables with a lot of text makes it difficult to identify the rows. Internal margins have been added to the columns. Additionally, they could be considered as figures or text boxes and put internal lines.

Reviewer 3 Report
This study is very interesting to read and holds a lot of promise. Overall, I think you did justice to elevating the key points through descriptions and poignant quotes, giving voice to the themes. I commented directly on a PDF version of the manuscript so my comments below are broader comments.
Areas that need to be addressed include:
- The description of Gademerian hermeneutical phenomenology, as developed by Fleming is not clear to me, as someone who has not used that methodology before. There are terms and words that are not widely understood or used in qualitative research methods used throughout 2.1 and 2.4.
- It is not clear how many researchers were actually involved and in what way. Are all the researchers co-authors in the paper? If so, why not use their initials in the body of the text?
- It will be important to have someone carefully proofread the next iteration as there are quite a few grammar errors throughout the paper; the present and past tense is inconsistent; there extra spaces and missing punctuations; and awkwardly phrases statements that need more clarity.
- Throughout the paper, you mix and match the terms “men”, “fathers”, “mothers”, and “women”. I would encourage you to stick to one or the other. At the very least, when referencing two different genders in the same sentence, please use the same form (e.g. mother and father, or women and men).
- The conclusion repeats, almost verbatim, parts of the paper. Please consider re-wording the conclusion with original sentences.
- I think the conclusion would be strengthened with more robust and explicit considerations for public health practices and policies. It is rather thin at the moment. It would be good to address more of the “so what?”
- This paper assumes heteronormative constructions/definitions of family and gender roles. Even if the participants fit a heteronormative nuclear family model, I think it may be important to acknowledge somewhere (perhaps limitations section) that the participants did not include single/lone parent participants, couples in non-hetero relationships, etc.

Author Response
This study is very interesting to read and holds a lot of promise. Overall, I think you did justice to elevating the key points through descriptions and poignant quotes, giving voice to the themes. I commented directly on a PDF version of the manuscript so my comments below are broader comments.
Response. Thank you for your constructive feedback, which we strongly believe has helped to improve our paper towards a potential publication. All your comments have been addressed and the changes made in the manuscript have been marked in red.
Areas that need to be addressed include:
The description of Gademerian hermeneutical phenomenology, as developed by Fleming is not clear to me, as someone who has not used that methodology before. There are terms and words that are not widely understood or used in qualitative research methods used throughout 2.1 and 2.4.
Response. We have rewritten section 2.1. Section 2.4 has also been modified to better explain the phenomenology developed by Fleming and how it has been implemented by the researchers.
It is not clear how many researchers were actually involved and in what way. Are all the researchers co-authors in the paper? If so, why not use their initials in the body of the text?
Response. Initials of the researchers, all co-authors, have been incorporated into the body of the text, indicating the contribution of each co-author.
It will be important to have someone carefully proofread the next iteration as there are quite a few grammar errors throughout the paper; the present and past tense is inconsistent; there extra spaces and missing punctuations; and awkwardly phrases statements that need more clarity.
Response. Thanks to the reviewers for taking the time to make numerous annotations on the pdf document. We have incorporated all these suggested changes and have asked a different independent native English-speaker to revise the manuscript. Changes made to the manuscript are marked in red.
Throughout the paper, you mix and match the terms “men”, “fathers”, “mothers”, and “women”. I would encourage you to stick to one or the other. At the very least, when referencing two different genders in the same sentence, please use the same form (e.g. mother and father, or women and men).
Response. Thanks for your suggestion. We have followed your advice and have unified all these terms using only “mothers and fathers”.
The conclusion repeats, almost verbatim, parts of the paper. Please consider re-wording the conclusion with original sentences.
Response. We have rewritten the conclusions for the most part:
PD affects family dynamics, including couple relationships. This can be strengthened by coping with adversity together or weakened by increased arguments or decreased sexual activity. Siblings and grandparents also suffer the consequences of PD in the family. Feelings of abandonment or overprotection may arise in older siblings and expressions of grief or even an opportunity to resolve past grief in grandparents.
Regarding their social environment, parents who suffer a loss due to PD experience work problems, lack of recognition of their grief, absence of rituals, trivialization of their loss and disavowal or delegitimization of their mourning.
I think the conclusion would be strengthened with more robust and explicit considerations for public health practices and policies. It is rather thin at the moment. It would be good to address more of the “so what?”
Response. We have added more specific implications for practice when looking after parents who suffer a perinatal death.
Policies that give the deceased foetus or baby the right to be registered and to have an identity or name should be developed. Religious rites should be offered in hospitals and communities for those parents who wish to celebrate them. At a community-care level, nurses, social workers, occupational therapists or psychologists should offer specifically targeted interventions to foster a healthy grief amongst families who have suffered a perinatal loss.
This paper assumes heteronormative constructions/definitions of family and gender roles. Even if the participants fit a heteronormative nuclear family model, I think it may be important to acknowledge somewhere (perhaps limitations section) that the participants did not include single/lone parent participants, couples in non-hetero relationships, etc.
Response. We have included this as a limitation.
All participants were part of either a nuclear or an extended nuclear family (with grandparents). None of the participants were neither single parents, nor couples in non-heterosexual relationships. Interviewing participants from single-parent families (or any other type of non-normative families) could have provided complementary results.
PDF comments:
We have addressed all the comments in the pdf
Thanks very much.
